# Advances in Climbing Robots for Vertical Structures in the Past Decade: A Review

**DOI:** 10.3390/biomimetics8010047

**Published:** 2023-01-22

**Authors:** Guisheng Fang, Jinfeng Cheng

**Affiliations:** 1College of Mechanical and Automotive Engineering, Zhejiang University of Water Resources and Electric Power, Hangzhou 310018, China; 2School of Electronic Information, Hangzhou Dianzi University, Hangzhou 310018, China

**Keywords:** vertical structure, climbing robot, application fields, adhesion mechanism, locomotion mode, control mode, operation tools

## Abstract

Climbing robots are designed to conduct tasks that may be dangerous for humans working at height. In addition to improving safety, they can also increase task efficiency and reduce labor costs. They are widely used for bridge inspection, high-rise building cleaning, fruit picking, high-altitude rescue, and military reconnaissance. In addition to climbing, these robots need to carry tools to complete their tasks. Hence, their design and development are more challenging than those of most other robots. This paper analyzes and compares the past decade’s design and development of climbing robots that can ascend vertical structures such as rods, cables, walls, and trees. Firstly, the main research fields and basic design requirements of climbing robots are introduced, and then the advantages and disadvantages of six key technologies are summarized, namely, conceptual design, adhesion methods, locomotion modes, safety mechanisms, control methods, and operational tools. Finally, the remaining challenges in research on climbing robots are briefly discussed and future research directions are highlighted. This paper provides a scientific reference for researchers engaged in the study of climbing robots.

## 1. Introduction

Climbing robots can replace human workers in tasks where they are required to climb along vertical or near-vertical objects. They can carry tools to conduct a wide range of hazardous tasks, such as detection, monitoring, cleaning, maintenance, installation, spraying, fruit picking, pruning, search and rescue, and so on. They are widely used in bridges, ships, chimneys, pipelines, streetlamps, nuclear power plants, wind power generation, high-rise buildings, agricultural picking, and other fields.

Since the first climbing robot, Mod-I, was developed by Nishi et al. [1] in the 1960s, climbing robots have attracted the attention of many research institutions and scholars. A large number of scientific research achievements and robot prototypes have emerged. In the past decades, scholars from all over the world have summarized the climbing robots made for use in different fields. Yun et al. [2] discussed the development status of bridge-cable-climbing detection robots. Megalingam et al. [3] summarized the technologies related to coconut-tree-climbing robots. Solanki et al. [4] elaborated on two key technologies of wall-climbing robots—the attachment method and motion mechanism. Fang et al. [5] reviewed the research progress of three different motion modes of wall-climbing robots: wheeled, crawler, and legged. In addition, they summarized four different adsorption technologies used in wall-climbing robots: negative-pressure adsorption, magnetic adsorption, bionic adsorption, and electrostatic adsorption. Seo et al. [6] summarized the climbing mechanisms, cleaning methods, and applications of robots used to clean the glass and facades of high-rise buildings. Cai et al. [7] and Hou et al. [8] discussed the research status of robots used for high-rise buildings and for defect detection on bridge cable surfaces, respectively. Bogue et al. [9] discussed the research status and potential applications of climbing robots. The above papers discussed the research status of climbing robots from the perspective of one or several application fields or key technologies. This paper focuses on a discussion of the six key technologies of climbing robots used for various vertical structures: their conceptual design, adhesion mechanisms, locomotion modes, safe-landing methods, control modes, and operation modes.

## 2. Overview of Research on Climbing Robots Used for Vertical Structures

The main types of climbing robots used for vertical structures are pole-climbing robots, pipe-climbing robots, tree-climbing robots, wall-climbing robots, cable-climbing robots, and robots that climb other irregular objects.

### 2.1. Pole Climbing Robot

Poles or tubes are widely used in streetlamps, lightning rod poles, building pipelines, and other structures. Most of these are cylindrical structures with diameters of 10–500 mm and lengths of a few meters to tens of meters. Some are variable-diameter structures, being wider at the bottom than at the top. The overall surfaces of the poles or tubes are smooth, some of which have steps and bending states. Pole-climbing robots are mainly used for surface detection, cleaning, and spraying of poles or tubes, as well as the maintenance of objects on poles.

With respect to pipe-climbing robots, Guan et al. [10,11] developed a truss-climbing robot called Climbot. The robot is composed of five single-degree-of-freedom joints and two claws and can climb truss structures and change lightbulbs. Noohi et al. [12] designed a pole-climbing robot called UT-PCR for the cleaning and maintenance of highway lamps. The robot consists of a triangular trunk and six mechanical arms with rubber wheels. In the climbing process, the robot can not only correct its deviations automatically but also cross a certain height of steps. Han et al. [13] designed a climbing robot for the nondestructive testing (NDT) of large pipe structures. The robot is composed of two driving modules and a driving connecting arm, and can cross external obstacles such as fixtures, flanges, and valves, as well as pipe components such as elbows and T-branch joints. Unlike robots that climb pipes from the outside, Agarwal et al. [14] designed a robot that can climb vertically within a pipe. The robot is composed of three symmetrically arranged track modules, which have two-way movement and can pass through internal complex T-shaped pipes and elbow networks at different angles. Verma et al. [15] developed a pneumatic-driven pipe-climbing soft robot. The robot is composed of a buckling pneumatic actuator and two pressure-drive rings. It can maintain climbing and cleaning performance even in wet conditions and underwater.

This review indicates that most pole-climbing robots are still in the stage of laboratory research and can only climb straight rods or tubes. Some robots can climb elbows and have a certain ability to negotiate obstacles and adapt to variable pipe diameters. A few robots have simple operation abilities and certain practical value. The adhesion of these robots is mostly achieved by clamping, while their locomotion is mostly of the inchworm type or wheeled type.

### 2.2. Tree-Climbing Robots

Trees are very different from rods and tubes as they have bark and branches, irregular shapes, and mostly uneven surfaces. In addition, their diameters can vary greatly, ranging from a few centimeters to more than ten meters. Tree-climbing robots can replace human workers in dangerous tasks such as pruning branches, picking fruit, pest elimination, and biological observation.

Lam et al. [16,17,18,19,20] developed a flexible tree-climbing robot, dubbed Treebot. It adopts innovative omnidirectional tree grippers and a continuum maneuver structure, and thus can adapt to a variety of tree species and achieve free switching from trunks to branches. It can be used to help workers with tree cultivation and biological observation. Ishigure et al. [21] developed the tree-pruning robot. It is composed of an up–down climbing mechanism, a steering mechanism, a posture adjustment mechanism, a chain saw mechanism, and a controller. Relying on self-weight and an energy-saving chainsaw drive, the robot can climb trees and prune them with low power consumption. Diller et al. [22] developed a tree-climbing robot named DIGbot, which can climb tree trunks. The robot consists of a body and six legs. Using a hook and claw installed on the legs, the robot can climb in all directions on rough trunks and can perform turns. Wibowo et al. [23] developed a coconut-harvesting robot, which adopts the spring-clamping and wheel-climbing methods. It can climb coconut trees with different diameters and carry cameras and blades to detect and cut down coconuts. Fu et al. [24] developed a robot for the pruning of fast-growing forests, which is composed of a wheel-climbing mechanism, a spring-clamping mechanism, and a ring-pruning mechanism. It weighs about 40 kg and can climb trunks with diameters of 150–350 mm at a speed of 20 mm/s, and can cut branches with a maximum diameter of 30 mm. Wright et al. [25] developed a multi-module snake-like tree-climbing robot dubbed Uncle Sam. The robot uses the spiral method to achieve the climbing movement, with a body that is wound around the trunk in a spiral. Upward and downward climbing movement is realized through synchronous rolling of the body.

This review of tree-climbing robots indicates that most robots can only climb straight trunks, while a few can transfer between trunks and branches and carry tools. Their adhesion modes are mostly clamping or claw stabbing, and their locomotion modes are mostly by wheels or tracks.

### 2.3. Cable-Climbing Robots

Cables and wire ropes are widely used in bridges, ships, cableways, hoisting machinery, and other scenarios. To ensure normal equipment operation, these cables need regular inspection, maintenance, and repair. The difference between a cable and a wire rope is that the periphery of a wire rope generally has no protective layers, while most cables do. A wire rope without a protective layer has a spiral shape and a certain flexibility. In contrast, a cable with a protective layer is cylindrical and more similar to a rod or tube. However, the protective layer is relatively soft compared with a rod or tube, and so it can be easily damaged while the robot is climbing. In addition, for suspension bridges or cable-stayed bridges, the cables are generally tens of meters or even hundreds of meters long, and some protective layers have cracks or bulges.

Ding et al. developed four generations of cable-climbing robots: CCRobot-I [26], CCRobot-II [27], CCRobot-III [28], and CCRobot-IV [29,30], as shown in Figure 1a–d. CCRobot-I is composed of a clamping module and a parallel manipulator. It weighs about 15 kg, has a load capacity of more than 30 kg, and its maximum climbing speed can reach >3 m/min. CCRobot-II adopts a palm-based grasping module and an alternate sliding frame mechanism, so its climbing speed can reach 5.2 m/min. It has a mass of 25 kg and maximum payload of 30 kg. To further improve climbing speed and payload capacity, CCRobot-III adopts a split-wire-driven method, being composed of a climbing precursor and a main frame. These two parts are connected and driven by steel wires. The climbing precursor acts as a moving anchor and moves quickly on the bridge cable. The main frame acts as a mobile winch, carrying the payload and pulling itself to a specific position with steel wires. It can climb at a speed of 12 m/min and can carry a load of more than 40 kg. The structure of CCRobot-IV is similar to that of CCRobot-III; it also consists of a climbing precursor and a payload-carrying body. The difference is that the new climbing precursor is replaced by a quad-ducted propeller-driven climbing system. CCRobot-IV can maintain a climbing speed and an optimal turning behavior of 12 m/min with a 40 kg payload. Its maximum climbing speed can reach 20 m/min. Wang et al. [31] designed the wheeled cable-climbing robot shown in Figure 1e. The robot is connected by two separate car modules through four turnbuckles to form a closed structure. The robot has a self-weight of 12 kg, can carry a maximum load of 8 kg, and can overcome obstacles 2.42 mm high. Xu and Wang et al. [32,33,34,35,36] designed a series of cable-climbing robots. The first generation of the wheeled cable-climbing robot, Model-I, is shown in Figure 1f. It is composed of three equidistant circular modules, which are connected by six connecting plates to form a closed hexagonal body for clamping cables. Each module includes two wheels for climbing, a CCD camera for visual inspection, two pairs of driving permanent magnets, and five Hall sensors for detecting magnetic leakage. It can perform defect-detection tasks on cable-stayed bridge cables. Subsequently, the project team designed an improved Model-II robot composed of two equally spaced modules connected by rods to form a closed hexagonal body that is fixed on the cable (Figure 1g). With the aim of building a robot able to detect broken wires within a spiral cable, the project team developed the Model-III robot in 2014 (Figure 1h), which is composed of a driving car and upper and lower support rods. The driving trolley and supporting connecting rods are connected through a fixed joint and installed relative to each other along the cable. A climbing device is installed on the body of the robot, which allows the car to rotate freely to adapt to guidelines with different pitches on the cables. In 2019, Xu et al. made further improvements by increasing the flexibility of the wheels via an extension spring and swingarm to achieve an obstacle-climbing function (Figure 1i). In view of the difficulties in detecting and repairing damaged bridge cables, Xu et al. designed the Model-IV cable-climbing robot based on independent quadrilateral suspension in 2021 (Figure 1j). The robot can automatically repair damaged bridge cables using testing, grinding, cleaning, spraying, and winding devices.

Cho et al. [37,38,39,40] designed three wire-rope-climbing and detection robots, named WRC^2^IN-I, WRC^2^IN-I+, and WRC^2^IN-II. WRC^2^IN-I is composed of a wheel-drive mechanism, an attachment mechanism, and a safe-landing mechanism. It can climb at 0.05 m/s with a 15 kg load. When a wheeled cable-climbing robot moves on an uneven cable surface, it produces periodic vibrations that affect detection accuracy. Therefore, the project team improved the first-generation robot by changing its wheeled structure into a tracked structure, which greatly reduces the vibration problem. To further simplify the installation and disassembly processes of the first-generation robot and improve its work efficiency, the project team developed the second-generation cable-climbing robot WRC^2^IN-II. The robot is composed of two separable attachment modules, two driving modules, and two obstacle-surmounting sub-modules. The improved robot can carry a load of 24 kg, while its installation and disassembly time is only about 5 min. Sun et al. [41] designed a wire-rope-climbing robot for detecting lamps at the top of streetlights at airports (Figure 2a). It is composed of a compression mechanism, a suspension mechanism, and a tracked movement mechanism. Its weight is 16 kg, and it can carry a 58 kg load. Ratanghayra et al. [42] designed a simple rope-climbing robot composed of a mounting frame and four mutually staggered wheels with motors. The wheels are pressed onto the rope by springs and can adapt to ropes of different diameters. Fang et al. [43] designed a six-wheeled wire rope-climbing robot called WRR-II for the maintenance of sluice wire ropes (Figure 2b). The developed climbing robot is composed of separable driving and driven trolleys. It adopts the spring clamping mechanism and the wheeled movement method. It can carry a camera and a laser-cleaning device to detect and clean sluice wire ropes.

This review of cable-climbing robots indicates that most research focuses on cable safety detection in cable-stayed bridges. A few are used for climbing wire ropes and soft ropes. The attachment modes are mostly clamping-type, while their movement modes are mostly wheeled-type and tracked-type.

### 2.4. Wall-Climbing Robots

Wall-climbing robots are widely used in the construction, shipbuilding, chemical, military, fire protection, and service industries, among others. They have become a focus of research on climbing robots, and hundreds of prototype systems have emerged so far. Compared with rods, trees, and other objects, walls have a large area. Walls can be rough or smooth, and some also have grooves and bulges, which creates challenges in the design of wall-climbing robots.

Heredia et al. [44] designed a window-cleaning robot named Mantis. It adopts three connected vacuum cup adsorption modules and a crawler movement mode. It uses translational and steering movement, while also independently crossing panes for glass cleaning. Bisht et al. [45] designed a robot for cleaning exterior glass walls that adopts the crawler movement mode and vacuum adsorption mode. It can carry a roller brush to clean glass curtain walls. Xiao et al. [46] designed a wall-climbing robot called the Rise-Rover, which has high reliability and strong load-bearing capacity. The robot adopts the pneumatic adsorption method and crawler-climbing method, and can quickly climb vertical walls with small grooves. Eto et al. [47] developed the WCR-Eto wall-climbing robot for hull welding, which uses a pair of two free rocker-arm hovering mechanisms with magnetic ball wheels to adapt to surfaces with a variety of shapes. It can cross 90° corners and 50 mm high obstacles. Milella et al. [48] and Eich et al. [49] developed crawler and wheeled climbing robots, respectively, for hull inspection tasks. Both robots use permanent magnets for adsorption and can perform real-time detection of hull defects with autonomous navigation. Seoul National University, South Korea, together with the Lingnan and Carnegie Mellon Universities, USA, developed four multi-connected crawler wall-climbing robots named MultiTank, FCR, Combot, and MultiTrack [50]. They use flat dry rubbers, rubber magnets, or suction cups as attachment devices, and all adopt the crawler drive mode. They can climb with a load in indoor, heavy industry, and building exterior wall scenarios, and have the ability to climb obstacles from plane to plane and from plane to circular arc. Souto et al. [51] developed a sandblasting robot for unsupervised automatic cleaning of large ships (Figure 3a), which adopts a separable double-frame structure to achieve alternating translation and rotation. Alkala et al. [52] designed a climbing robot named EJBot for the needs of petrochemical container detection (Figure 3b). It is composed of a propeller-drive unit, a wheel-drive unit, and a wireless control unit. It can adapt to climbing and detection tasks on a variety of surfaces with different materials and bending degrees and can cross 40 mm high obstacles. Lee et al. [53] designed a modular wall-climbing cleaning robot that can surmount obstacles for the cleaning of exterior glass walls of buildings. The robot is composed of a main platform and three independently scalable modular climbing units. The robot uses a winch at the top of a building to move up and down and uses an air pressure adsorption device and cleaning device in its middle module to bring the robot close to the wall for cleaning tasks. Each module is equipped with sensors to detect obstacles and walls so that the robot can automatically avoid obstacles.

This review of wall-climbing robots indicates that most current research focuses on tasks related to buildings and hulls. The attachment modes are mostly vacuum or magnetic adhesion, and the moving modes are mostly tracked-type or wheeled-type.

### 2.5. Climbing Robots for Other Irregular Vertical Structures

In addition to pole-, tree-, and wall-climbing robots, climbing robots have been designed for performing detection and maintenance tasks on irregularly shaped vertical structures. For example, there are robots for steel bridge climbing, tower climbing, wind turbine blade climbing, and cloth climbing.

Among steel bridge detection robots, Nguyen and La et al. [54,55,56,57,58,59,60] designed the crawler and hybrid climbing robots. The crawler-climbing robot uses a reciprocating mechanism and a roller chain, which enables it to climb on structures with different shapes and from one surface to another. The hybrid climbing robot uses a combination of wheels and legs for climbing. On the smooth surfaces of a steel bridge, it can use the wheels to move quickly. When it needs to cross obstacles or realize plane conversion, it can use its legs. Pagano et al. [61] designed a seven-degree-of-freedom (7-DOF) inchworm-like climbing robot, and adopted a real-time path planning method based on the LOS algorithm so that the robot can climb autonomously in restricted areas within steel bridges. Wang et al. [62] designed a four-wheel climbing robot composed of a body, four magnetic wheels, a steering system, and a shock absorber. The robot can climb vertical surfaces and reverse horizontal surfaces and can cross complex obstacles, such as bolts, steps, convex corners, and concave corners. Ward et al. [63] designed an inchworm-like climbing robot called CROC, which consists of a seven-DOF trunk and two magnetic foot pads. Each magnetic foot pad includes three independently controlled magnetic toes. The robot can perform 360° plane conversion and pass through manholes.

For the inspection and maintenance of transmission towers, Lu et al. developed two climbing robots dubbed Pylon-Climber I [64] and Pylon-Climber II [65]. Both robots use gripper adhesion and step-by-step driving. They can climb straight angle irons, cross between angle irons, and climb over obstacles such as bolts. Compared with Pylon-Climber I, Pylon-Climber II has improvements in its clamping jaw design. Instead of clamping the entire angle iron, it only clamps a single side, making its structure simpler and more efficient. Yao et al. [66] designed a series-parallel hybrid transmission-tower-climbing robot composed of two parallel legs with 3-DOF delta mechanisms and a trunk linkage mechanism. The legs are equipped with electromagnets, which can be adsorbed onto the transmission tower. Relying on inchworm gait control, it can achieve climbing and obstacle negotiation functions.

Lee et al. [67] designed a climbing robot for the maintenance of offshore wind turbines. It has a rectangular frame structure composed of four risers, two grippers, two operating arms, and a mobile scissor device. The robot can climb towers or blades and performs cleaning and inspection using waterjets and phased array ultrasonic testing (PAUT) devices, respectively. Birkmeyer et al. [68] developed a robot called CLASH that can climb loose vertical cloth. Liu et al. [69] also developed a soft-cloth-climbing robot named Clothbot. It uses two wheel-shaped clamping claws to clamp onto the wrinkles of clothes, and uses a 2-DOF omnidirectional tail to adjust the center of the robot so that it can maintain its balance and change its rotation direction.

Designing robots to climb irregular objects with highly variable structures and shapes is difficult, and general adhesion mechanisms and locomotion modes remain lacking.

## 3. Basic Design Requirements of Climbing Robots for Vertical Structures

Climbing robots are mainly used to carry out risky tasks in hazardous environments, so they require certain basic characteristics, such as functionality, a light weight, strong load-carrying capacity, flexible movement, a fast climbing speed, high safety, strong environmental adaptability, and the ability to climb objects without damaging their surfaces, as detailed below:

(1) Functionality. This is the primary consideration in climbing robot design. Each has a purpose, such as detection, cleaning, spraying, installation, or maintenance. Therefore, in addition to a basic climbing ability, climbing robots also need to have a certain load-carrying ability, such as the ability to carry a camera or a nondestructive testing device for defect detection, cleaning equipment, or a manipulator for installation and disassembly tasks.

(2) Lightweight structure. Climbing robots should be as light as possible to minimize their size and energy consumption.

(3) Fast climbing speed. Robots are mainly used to replace skilled workers to conduct tasks that are difficult, hazardous, or boring, so their work efficiency must be higher than that of humans.

(4) Good environmental adaptability. The objects that need to be climbed are diverse and have different shapes, so climbing robots need good environmental adaptability to be able to climb objects of various diameters, lengths, materials, shapes, tilt angles, and surface roughnesses.

(5) Obstacle negotiation ability. The surfaces of climbed objects are not always flat and smooth. Some have bulges, pits, steps, or forks, which requires robots to have good obstacle negotiation abilities.

(6) Working safely and reliably. During the climbing process, a robot can experience a power failure, jammed mechanism, or other fault. Working at heights can cause vibrations and shaking due to wind, which can affect the safety of robots and their operators. This requires robots to have a self-protection ability so that they will not fall from height in the case of a power failure or can be safely recovered in case of jamming.

(7) A general installment interface. Climbing robots should have a general installment interface and carry multiple tools that can be changed as the robot is working to expand its functional range. In addition, the impact of the tools on the robot’s performance should be minimized.

(8) Other factors to be considered include structural size, cost, energy supply mode, and installation and disassembly times. Some working spaces are limited, necessitating a small robot. Robots powered by a cable may be unsuitable for work on long objects. In addition, manufacturing costs must be considered; accordingly, components and modules that can be bought online should be preferred.

## 4. Key Technologies Used in Climbing Robots

### 4.1. Conceptual Design of Climbing Robots

Conceptual design is an early stage in the product design process that has an essential effect on robot innovation. The conceptual design of robot products describes the combination of principal components used in the space or structure required to meet the customers’ functional requirements. Once the conceptual design is completed, 60–70% of the product design is determined. Therefore, conceptual design is very important and is key to distinguishing between products.

For climbing, most robots adopt a conceptual design with conventional structural shapes, such as rectangular structures, triangular structures, hexagonal structures, and circular structures. For example, the tree-climbing robot designed by Gui et al. [70] adopts a triangular structure, being composed of three symmetrically distributed wheel mechanisms. The cable-climbing robot designed by Xu et al. [33] adopts a rectangular structure, which is connected by two symmetrical modular trolleys.

Many animals have good climbing abilities, such as geckos, cockroaches, spiders, inchworms, sloths, monkeys, snakes, cats, and beetles. Inspired by animals, researchers have designed various bionic climbing robots. Wang et al. [71] designed a quadrupedal tree-climbing robot that mimics the tree-climbing movements and postures of monkeys. Liu et al. [72] designed a climbing robot that adopts a five-link mechanism and a piston mechanism to imitate the climbing movement of geckos. Bian et al. [73] designed a foldable climbing robot that imitates the attachment and climbing mechanisms of longicorns and geckos, as shown in Figure 4a. By imitating the attachment mechanism of cicadas and geckos and the gecko climbing gait, Bian et al. [74] designed a gear-and-link-driven climbing robot, as shown in Figure 4b. Kanada et al. [75] imitated the operating mechanism of leeches and designed a soft climbing robot called LEeCH. Yanagida et al. [76] designed a climbing robot named Scorpio, which imitates the crab spider species Cebrennus rechenbergi. Inspired by the behavior of arboreal snakes in climbing tree trunks, Liao et al. [77] designed a snake-like winding-pole-climbing soft robot. Han et al. [78] developed a caterpillar-inspired segmented robot that can climb vertical surfaces.

In addition to climbing biomimetics, researchers have also designed climbing robots that imitate the growth and climbing actions of plants. Fiorello et al. [79] provided an overview of the methodological approaches and tools exploited by researchers for extracting the relevant biological features of climbing plants that might be adapted to design the plant-inspired robotics under three main themes: adapation, movements, and behavior. Mazzolai et al. [80] offered a brief review of the fundamental aspects related to a bioengineering approach in plant-inspired robotics, including the movement mechanism of roots and the attachment and climbing mechanisms of shoots.

### 4.2. Adhesion Methods

Climbing robots often need to adhere to different vertical surfaces. Commonly used adhesion methods include magnetic adsorption, air pressure adsorption, clamping adhesion, claw grasping, electrostatic adsorption, and biological adhesion.

#### 4.2.1. Magnetic Adsorption

The magnetic adsorption method adopts a permanent magnet or electromagnet (or a combination) and is suitable for use with ferromagnetic objects.

Permanent magnet adsorption is one of the most common magnetic adsorption methods. It can be divided into contact and non-contact types according to whether the magnet is in contact with the surface of the climbed object. Contact permanent magnet adsorption involves a combination of a permanent magnet and a moving mechanism. For instance, Erbil et al. [81] adopted the magnetic wheel method in the PC-101 pole-climbing robot. Fourteen magnets are arranged on each wheel in the circumferential direction. As the wheel rotates, two or three magnets always act on the pole. The MIRA climbing robot designed by Ahmed et al. [82] is composed of a group of permanent magnets that are regularly arranged and embedded into a polyurethane wheel frame. Tavakoli et al. [83,84] designed four generations of magnetic omnidirectional wheels in the Omnilimbers climbing robot. The first generation of magnetic wheels adopted integral ring magnets, the second generation adopted a magnet array, the third generation featured an evenly arranged magnet array in the middle of an omnidirectional wheel, and the fourth generation adopted magnetic rollers.

Contact permanent magnet adsorption systems have a compact structure but cause wear during movement. Non-contact permanent magnet adsorption systems have separate mechanisms for adsorption and movement, leaving a gap between the adsorption device and the surface. For example, Howlader et al. [85] used a non-contact permanent magnet adsorption mechanism in a reinforced-concrete wall-climbing robot. The robot is composed of a mobile platform, four wheels, and a magnetic adsorption module fixed under the mobile platform. The magnetic adsorption module is 2 mm away from the wall, with a yoke and 3 N42 neodymium magnets arranged in the N-S-N direction, which significantly increases its adsorption force. Yan et al. [86] adopted a multi-directional magnetized permanent magnet adsorption device (PMAD) with adjustable spacing in a climbing detection robot for a hydropower station. The magnetic adsorption device is composed of multiple groups of permanent magnet arrays and a magnetizable base. The robot adjusts the position of the PMAD through a connecting rod and screw pair mechanism to dynamically adjust its adsorption force. Ding et al. [87] adopted a non-contact permanent magnet adsorption device with surface adaptability in a wall-climbing robot developed for ultrasonic weld detection in spherical tanks. There is a gap of 5–8 mm between the magnetic device and the surface. This scheme not only achieves higher magnetic energy utilization than magnetic tracks or magnetic wheels but also has the flexibility of magnetic wheel technology. To ensure a stable adsorption force for robots climbing irregular or large-radius surfaces, Silva et al. [88] adopted an adsorption device that can dynamically adjust the position of the permanent magnet. The device uses two inductive sensors to maintain a constant distance between the magnet and the climbed surface via a worm-drive shaft to keep the adsorption force stable.

In permanent magnet adsorption, the magnetic force is fixed. While energy is not required to maintain the magnetic force, it cannot be turned off. To solve this problem, switchable permanent magnets have been used. For example, Tavakoli et al. [89] used switchable MagJig 95 magnets in the Omnilumber-II climbing robot. Using a handle connected to the top of the robot, the user can manually rotate the moving magnet to switch it on and off. However, this device can only switch the magnetic force on or off and cannot adjust its strength.

Electromagnetic adsorption uses the electromagnetic principle to energize an internal coil to generate a magnetic force. Electromagnetic adsorption has also been used in climbing robots because it can be used to switch the magnetic force on and off as well as adjust its strength. For example, Minibobot [90] uses two electromagnetic feet with an alternating adsorption action for climbing. Han et al. [91] designed a distributed electromagnetic adsorption device in a hull-rust-removal robot. The robot uses a double-chain crawler as the mobile device, with an electromagnetic adsorption device installed on a track, allowing it to move with the crawler. According to the electromagnetic adsorption principle and control requirements, a distributed control mode is used to accurately adjust the magnetic force of each part of the adsorption module.

In addition to permanent magnetic adsorption and electromagnetic adsorption, hybrid magnetic adsorption devices based on their advantages have been designed. For example, Cardenas et al. [92] designed a magnetic wheel that uses electro-permanent-magnet (EPM) adsorption technology. The magnetic wheel is composed of two permanent magnets with different magnetism (such as neodymium and Alnico5), two magnetic poles, and copper-enameled coils. EPM allows the magnetic force to be controlled by simply applying a short electrical pulse to the coil winding. By controlling the amplitude of the electric pulse, the magnetic force can be adjusted to the required value to realize continuous changes in magnetic adhesion.

The main advantage of magnetic adsorption methods is that their adsorption force is strong. Permanent magnets do not need additional energy, or only a small amount. Electromagnets can control the magnetism by switching on and off. The disadvantage of magnetic adsorption is that it is not applicable to non-ferromagnetic materials such as cement, brick, or stainless steel. Furthermore, some applications need to be electromagnetic-proof and explosion-proof. The magnetic adsorption force is related to the area of the magnet and the distance between the magnet and the metal surface. Its strength decays rapidly with distance from the object’s surface. Magnets are generally heavy, which increases the weight of the robot and reduces its load-carrying capacity. For permanent magnets, the magnetic force is fixed and difficult to eliminate. Electromagnets need an uninterrupted power supply. When power is lost, the magnetic force will disappear, posing certain safety risks.

#### 4.2.2. Air Pressure Adsorption

On large and flat surfaces, climbing robots often use air pressure adsorption, which may be active or passive. Active adsorption uses a vacuum, negative pressure, or aerodynamic adsorption. Passive adsorption uses suction cups without additional vacuum pumps or negative-pressure chambers.

Vacuum suction is the most common air pressure adsorption method. It exhausts the air from a suction cup using a vacuum pump to adhere to the climbed surface. Guan et al. [93] used a vacuum adsorption device in a bipedal wall-climbing robot called W-Climbot. The adsorption device consists of three suction cups, a support plate, a dry rotary vacuum pump, and some accessories. Three cups are mounted on the vertices of the equilateral triangle of the support plate, which can reduce the tilt of the robot caused by the deformation of the suction cups. Pressure sensors are installed in the suction cups to measure the vacuum inside the cup and output it to a low-level controller in real time to achieve closed-loop pressure control and a stable adsorption force. Vacuum adsorption is easy to control and has a high load capacity. It is not limited by the surface material but by its quality and is generally suitable for smooth planar objects. The strength of its adsorption force is related to the pressure difference and adsorption area. If there are holes or gaps in the surface, the adsorption force will be greatly reduced. In addition, vacuum adsorption requires a vacuum pump and a good sealing chamber, which increase the energy consumption, weight, and noise level of the robot.

Negative-pressure adsorption uses the adsorption force generated by an impeller or eddy current to fix the robot to a surface. An eddy current can cause local negative pressure via a rapidly spirally rotating airflow in a closed cavity, which is somewhat similar to the tornado effect. For example, Zhao et al. [94] adopted the eddy current adsorption method in the Vortexbot wall-climbing robot. The adsorption mechanism consists of a vortex ring, an annular skirt, an upper cover, and four symmetrically distributed nozzles. Airflow through the four nozzles creates negative pressure in the vortex chamber, which presses the robot against the surface. Eddy current adsorption does not require suction cups, so it can adapt to rough surfaces and obstacles. The negative-pressure adsorption method was adopted in the LARVA wall-climbing robot developed by Koo et al. [95]. The adsorption mechanism consists of a vacuum chamber, an impeller with a motor, and a double-layer sealing device. When the motor drives the impeller, the air in the vacuum chamber is expelled, creating a pressure difference between the environment and the vacuum chamber, so that the robot adheres to the surface. Parween et al. [96] used two negative-pressure adsorption modules in the Ibex climbing robot. Each adsorption module consists of a main suction chamber, suction cups, impellers, and connectors. Through rotation of the impeller, a pressure difference is generated between the main suction chamber cavity and the atmospheric pressure. The suction cup is equipped with a skirt that maintains the pressure differential in the chamber and creates a positive force that keeps the suction cup attached to the surface.

Aerodynamic adsorption uses the wind generated by a propeller to attach a robot to a surface, allowing it to adapt to surfaces of various shapes. Faisal et al. [97] developed a wall-climbing robot that uses the air pressure difference thrust generated by two ducted fans to attach the robot to a wall. Sukvichai et al. [98] used a double-propeller attachment mechanism in a wall-climbing robot. The two propellers have the same structures with opposite directions of rotation and can be controlled by servo motors for angle adjustment, so that the robot can achieve two-wheel attachment and four-wheel attachment according to the climbing conditions. Mahmood et al. [99] adopted a propeller-type attachment mechanism in the UOTWCR-II wall-climbing robot. It consists of two left-hand and right-hand rotor systems and two drive wheels. Another front steering wheel is connected to the structure to support the robot. The left rotor has a clockwise thruster, while the right rotor has a counterclockwise thruster. The two rotors rotate in different directions, creating a downward thrust that keeps the robot attached to a surface.

Passive adsorption uses multiple suction cups to alternately engage and disengage, so that the robot can attach to a flat surface. Passive adsorption systems are lightweight and quiet because they do not require a vacuum pump or negative-pressure device. They have been used in the design of plane-climbing robots. For example, Ge et al. [100,101] used a passive suction cup structure in a smooth-wall-climbing robot. Multiple passive suction cups are fixed on the outer surface of the crawler at equal intervals and rotate with the crawler. Under the action of a guide rail, they can be attached to the wall and then pressed and separated.

In addition to conventional pneumatic adsorption methods, some new methods have emerged, such as vibration adsorption. Chen et al. [102] installed a vibration adsorption device on the feet of a gecko-shaped wall-climbing robot. The adsorption device consists of four parts: a vibrating mechanism, an air-releasing mechanism, a guiding mechanism, and a stability retainer. The vibration mechanism generates periodic vibrations so that the suction cup on it can generate a stable negative pressure and adsorb on the surface. The air release mechanism can quickly release the module from the wall when it is not working. The guide mechanism is used to move the vibration mechanism up and down linearly. Stability retainers are used to prevent unexpected vibrations of the robot body. Compared with the suction cups used in conventional structures, the vibration adsorption method can obtain a stronger and more stable adsorption force, and its environmental adaptability is also better.

To improve the adsorption effect, air pressure adsorption systems can be combined. For example, in the Rise-Rover wall-climbing robot developed by Xiao et al. [45], vacuum adsorption and duct fans are used simultaneously so that the robot can adhere to smooth surfaces and also span grooves. Air pressure adsorption is unsuitable for use in space due to the absence of air.

#### 4.2.3. Clamping Adhesion

Clamping adhesion systems use grippers or other encircling mechanisms to attach a robot to a structure. According to the way the clamping force is generated, clamping methods can be divided into several forms, such as pneumatic clamping, electric clamping, spring clamping, mechanical clamping, and serpentine winding.

Electric clamping relies on the driving force generated by a motor to clamp jaws onto an object. For example, Tavakoli et al. [103] adopted a gripper structure in the 3DClimber pipe-climbing robot. The gripper consists of two unique multi-fingered V-jaws, a brushless DC motor, left and right ball screws, and two linear guides. Force sensors and strain gauges are installed on the jaws, which can sense their clamping force and deformation in real time. Chen et al. [104] adopted a humanoid embracing structure in the EVOC-1 climbing robot. The embracing device is composed of three joints, three link mechanisms, a torsion spring, and other components. The drive motor causes the push rod to push the root joint to make a circular motion around the frame of the driving part.

Spring clamping relies on the force of an adjusting nut and spring. For example, the WRC^2^IN-I robot [37] adopts a pantograph attachment device to bring it close to a steel bridge cable. The device consists of ball screws, pantographs, springs, wheels, brackets, ratchets, and handles, and is similar to the pantograph mechanism of a train. When the handle is rotated, it rotates a double-helix screw so that the left and right sliders on the screw move to both sides, and the attachment mechanism is brought close to the cable via the action of the connecting rod. The WRC^2^IN-II robot [40] uses a spring clamping method to attach to steel bridge cables. A handle is used to adjust the clamping force of the spring.

Mechanical clamping relies on the force of a mechanical structure. Sun et al. [41] used a pressing mechanism in a light-pole-climbing robot that consists of a handle, a wedge-shaped extrusion block, a clamping block, a connecting rod, and rubber teeth. When the handle is rotated, the wedge-shaped extrusion block is moved up and down by the thread at the front end of the handle so that the clamping block moves to the right and the rubber teeth clamp firmly to the wire rope.

The clamping attachment method can be easily adapted to slender rod-shaped objects, such as beams, columns, pipes, and trees. However, it is not suitable for flat objects.

#### 4.2.4. Claw Grasping Attachments

Insects and arthropods often use their thorny feet to climb natural or man-made structures. The claw grasping method is a bionic attachment method that uses a claw-thorn structure to anchor to the surfaces of relatively rough objects, such as brick walls, tree trunks, and rock walls.

The claw grasping method was firstly applied in the Spinybot climbing robot [105]. Spinybot uses cockroach-like barbed feet to climb hard, flat surfaces such as concrete and brick walls. Later, Haynes et al. [106] also adopted a similar barbed structure in the RiSE robots. The barbs allow climbing of both hard and soft objects, such as blankets and cork. Lynch et al. [107] adopted a RiSE-like claw-thorn structure in the DynoClimber wall-climbing robot. Ji et al. [108] developed a four-legged robot based on flexible pads with claws that has the ability to climb on rough vertical surfaces. Each pad consists of ten toes, each separated in a radial form. Lam et al. [17] used an omnidirectional claw-thorn mechanism in Treebot composed of four claws. Each claw consists of two phalanges. The tip of each phalanx is equipped with a sharp surgical suture needle, which can penetrate the object being climbed. Under the action of the linear motor and spring, a link mechanism is used to clamp and loosen the clamping jaw. The gripper has a wide curvature, can climbing various tree species, and can clamp the surface of an object using a spring without requiring electricity, providing a good energy-saving effect. Xu et al. [109] used a four-claw gripper with a cross structure in a climbing robot. Each gripper is composed of two pairs of small hooks with a certain elasticity, so that the gripper can grasp rough wall surfaces with improved stability. Liu et al. [110] designed a barbed, bipedal, wall-climbing robot by imitating a known barb structure. The robot has a pair of bionic, spiny, flexible claw feet. Each foot consists of two spiny claws, a spring, a servo, and a cam mechanism. The movement of the cam mechanism is controlled by a steering gear to realize clamping and loosening of the two claws. In the LEMUR robots developed by Parness et al. [111], a ring-shaped micro-thorn gripper is used. The ring gripper is composed of 16 finger-shaped thorns, which are designed in a layered structure to adapt to the surfaces of objects of different scales. Inspired by the micro-thorn structure of the LEMUR robots, Li et al. [112] designed a similar annular micro-thorn claw grasping mechanism. The claw structure consists of 160 flexible micro-thorns evenly distributed on 16 brackets. Liu et al. [113] adopted a barbed wheel in the Tbot wall-climbing robot. The robot consists of two driving wheels and a flexible tail. Each driving wheel is composed of eight layers of thorns, and a partition is installed between the connected thorns. Each wheel thorn piece contains four thorn claws connected to the wheel hub via flexible suspension. This structure enables Tbot to attach to rough walls and attain a high climbing speed. In the six-legged wall-climbing robot developed by Han et al. [114], twelve gripping spiny feet are used to allow the robot to crawl in any direction on a rough wall.

The claw grasping method can adapt well to rough surfaces and does not require power when static. Hence, it is energy-efficient but has difficulty adapting to particularly smooth surfaces such as glass.

#### 4.2.5. Adhesive Adsorption

Adhesive adsorption systems imitate climbing animals such as geckoes or tree frogs, and are divided into dry adhesion and wet adhesion systems. Dry adhesion relies on the van der Waals force between molecules to attach the robot to a surface. Wet adhesion relies on surface tension and the capillary and viscous forces between liquids to adhere a robot to the surface of a wet object.

Dry adhesion. Borijndakul et al. [115] briefly reviewed the characteristics of bio-inspired adhesive foot microstructures used on the climbing robots for smooth vertical surfaces, namely spatula-shaped feet and mushroom-shaped feet. Kalouche et al. [116] developed the ACROBOT climbing robot for inspecting equipment racks in the International Space Station. A synthetic gecko-like cushion is used to adhere to surfaces, which is composed of a suspension layer and an oriented adhesive layer. The suspension layer conforms to rough surfaces to compensate for small misalignments of the cushion. The oriented adhesive layer contacts the surface of the object to generate van der Waals forces. Murphy et al. [117] used a dry elastomer adhesive as the adhesion material in the first-generation Waalbot wall-climbing robot. However, dry elastomer adhesives lose their grip when contaminated, causing the robot to fall. Later, a great improvement was made in the second-generation Waalbot II. An imitation gecko-fiber-hair adhesive pad is used as the sole adhesion material, and a sticky autonomous recovery mechanism is used so that the robot can reliably adhere to smooth or near-smooth surfaces. In the first-generation spider-like robot Abigaille-I developed by Menon et al. [118], a synthetic dry adhesive pad is used as a foot pad to adsorb to smooth surfaces. The second-generation lightweight climbing robot Abigaille-II developed by Li et al. [119] adopted a plantar structure composed of adhesive patches. The plantar patch has microhairs with mushroom-like caps attached to the tops of millimeter-scale flexible posts, allowing them to adhere to smooth surfaces. Henrey et al. [120] used a layered dry adhesive as a foot-pad material in the third-generation hexapod bionic wall-climbing robot Abigaille-III. The layered dry adhesive is composed of a rigid substrate, a polydimethylsiloxane (PDMS) macro-pillar array, and a micro-pillar array. Three-layer materials are bonded by silica gel, which can attach to smooth and uneven surfaces. Yu et al. [121] developed a robot that can crawl stably on a flexible surface in microgravity with the help of gecko-inspired toe pads. The adhesive pads are based on the microstructure of the dry-adhesion polyvinyl siloxane (PVS) material. Liu et al. [122,123] used adhesive foot pads in the climbing robots AnyClimb and AnyClimb II. Wang et al. [124] adopted an attached foot pad based on thermoplastic adhesive (TPA) bonds in the ThermsBond climbing robot. The rheological properties of TPA give it a large payload capacity, making it useful for various flat surfaces and complex vertical terrain. Osswald et al. [125] used a hot-melt adhesive (HMA) technique to achieve a new type of autonomous robotic climbing. HMA is an economical solution to improving adhesion that acts by controlling the material temperature. The robot is equipped with servo motors and thermal controls to actively change the temperature of the material, and the coordination of these components allows the robot to walk against gravity at a relatively large bodyweight.

Wet adhesion. He et al. [126] designed and fabricated a combination of electroformed and soft-etching technology by analyzing the way that stick insects climb vertical surfaces using their smooth foot pads. A wet adhesion pad with a novel microstructure was applied to a prototype six-legged wall-climbing robot, and a good adhesion effect was achieved. In the climbing robot developed by Wiltsie et al. [127], a novel adhesion effect based on a magnetorheological fluid was used. Magnetorheological fluids are novel “active” or “smart” fluids composed of micron-sized iron particles suspended in an inert oil, and have controllable fluidity. They exhibit low-viscosity Newtonian fluid properties in the absence of an external magnetic field, while they act as a Bingham fluid with high viscosity and low fluidity in an external magnetic field. There is a corresponding relationship between the viscosity of the liquid and the magnetic flux. This conversion consumes low amounts of energy, is easy to control, and has a rapid response.

Adhesive adsorption methods, whether dry or wet, do not require an energy supply. Their disadvantage is that the adhesion force is small and, when the adhesive pad is contaminated, the adhesion effect is greatly reduced or absent, so these methods are unsuitable for outdoor use.

#### 4.2.6. Electrostatic Adsorption

Electrostatic adsorption methods cause the robot to adsorb to an uncharged surface based on the principle of electrostatic induction. Wang et al. [128] used electrostatic adsorption technology in a thin and flexible climbing robot designed for narrow gap detection in industrial equipment. The robot consists of a forefoot and a torso. The forefoot is composed of two short adsorption electrode films and a driving motor film, and the torso is composed of a long driving electrode film and a short adsorption electrode film. Wang et al. [129] proposed an inchworm-like robot composed of flexible printed circuit films for the inspection of narrow gaps in large machines such as generators in power plants. It uses electrostatic adhesion and electrostatic thin-film actuators to achieve a structure with low height. Li et al. [130] used an electrostatic attachment pad for adsorption in a crawler-type wall-climbing robot. It has the advantages of strong adaptability to various walls, a relatively light weight, and a simple structure. Gu et al. [131] used an electro-adhesion technology in a soft climbing robot to adsorb on the surfaces of wood, paper, glass, and other objects. Attachment and detachment of the robot are realized by adding and cutting off the voltage to the electro-adsorption feet. Electrostatic adsorption is suitable for smooth and clean surfaces, but the adsorption force is small and the load capacity is weak.

#### 4.2.7. Hybrid Adhesion

Hybrid adhesion methods use a variety of adhesion methods to enhance a robot’s adhesion abilities. Xu et al. [132] designed a wall- and glass-climbing robot that uses three attachment methods: micro-thorn grasping, viscous adsorption, and vacuum adsorption. When climbing on rough walls, the robot adopts micro-thorn grasping and vacuum adsorption methods, and when climbing on smooth glass surfaces, it adopts viscous adsorption and vacuum adsorption. Liu et al. [133] imitated the climbing and adsorption functions of flies and larval fish. The attachment device of the robot is composed of a grasping mechanism, an adhering mechanism, and an adsorption mechanism. The gripping mechanism consists of four ratchets for gripping particles on rough walls. The suction mechanism consists of a turbofan, suction cups, and flexible skirts, which can provide suction for the robot during the entire motion cycle. The adhesion mechanism consists of an adhesive material that provides adhesion to a wall surface. Inspired by the climbing strategy of geckos, Ko et al. [134] proposed a crawler motion solution that simultaneously uses static electricity, elastomer adhesion, and tail force in a climbing robot. Hybrid adsorption can be suitable for different surfaces, but the adsorption device requires a complicated structure.

#### 4.2.8. Other New Adhesion Methods

In recent years, with the development of new materials and technologies, some new adhesion methods have emerged. Huang et al. [135] developed a boronate polymer hydrogel and applied it to a climbing robot. It can rapidly switch between bonded and non-bonded states in response to mild electrical stimulation between 3 V and 4.5 V. William et al. [136] adhered a robot to surfaces by means of gas lubrication generated by vibration. This robot uses an eccentric rotor motor (ERM) as a suction device. This motor can drive a 14 cm diameter floppy disk to generate 200 Hz vibrations. With these vibrations, a low-pressure gas film with a thickness of several hundred microns is created between the robot and the surface, providing the robot with sufficient adhesion. Compared with other climbing robots, the robot is lighter in weight, lower in cost and power consumption, and has a great application space in high-altitude operations. Since these new attachment technologies are still in the process of exploration, their adhesion performance remains to be further verified.

All of the above adhesion methods have their advantages, disadvantages, and applicable scopes. A performance comparison is provided in Table 1.

### 4.3. Locomotion Modes

Locomotion mechanisms enable a robot to move up and down or side to side on the inner and outer surfaces of climbed objects. Locomotion modes can be divided into active and passive types. The passive type is generally rope-driven and mainly relies on external power, such as a hoist and winch, to move a robot via traction of a cable. The robot itself does not provide power. According to the movement mechanism, active types can be divided into wheeled, legged, tracked, inchworm, and hybrid types.

#### 4.3.1. Rope-Driven Locomotion

Rope-driven climbing robots are tethered to a rope that is pulled by a winch. Fujihira et al. [137] developed a rope-driven steel-cable-climbing robot for detecting cables in suspension bridges under strong wind conditions. The robot relies on the coordinated action of a wire rope and ascender to move up and down. Seo et al. [138,139] designed a parallel climbing robot that consists of a measuring device and two lifters. It can carry heavy loads for surface work in large workspaces. Lee et al. [67] designed a wire-rope-driven parallel robot system for offshore wind turbine maintenance. The robotic system consists of a mobile platform and two manipulators. The mobile platform sets four hoists at the four corners of the robotic system for the climbing of wind turbine towers or blades. Each hoist contains a pulley that controls the length of the wire between each hoist and the nacelle to achieve positional and orientation control of the mobile platform. Begey et al. [140] designed a three-cable-driven parallel robot called PiSaRo2, which consists of three cables, three pulleys, three winches, and an end effector. The robot can be raised or lowered via a winch. Seo et al. [141] designed a parallel robot driven by double hoisting cables. The robot consists of two lifters and two rope-measurement sensor structures. Unlike other cable-driven parallel robots that require an external winch structure, this robot requires only two ropes, as the traction pulleys of the hoist allow the robot to climb using the ropes. Rope-driven climbing robots can be large, carry heavy loads, and have good climbing speed, stability, and safety. Their disadvantage is the need for a winch-and-pulley lifting system, which reduces their flexibility and increases manufacturing and installation costs.

#### 4.3.2. Wheeled Locomotion

Wheeled climbing robots are inspired by automobiles. Wheels are one of the most common locomotion modes and have been applied to many climbing robots. Wheeled climbing robots commonly have three [94,142], four [52,62], or six wheels [12]. The cable-climbing WRC^2^IN-I robot adopts three sets of wheel-drive devices that are evenly distributed in the circumferential direction at 120° angles to each other. The drive device is composed of a DC brushless deceleration motor, a toothed clutch, a bevel gear, a spur gear, an arc wheel, and a support frame. In the power-on state, the clutch works, the motor drives the arc wheel to rotate through the bevel gear and spur gear, and the robot climbs using the friction between the wheel and cable. Zheng et al. [142] designed a lightweight wheeled cable-climbing robot composed of three-wheeled climbing modules enclosed by hinges. Two of the three modules are drive modules, while the other is a passive module. Each module is fitted with two wheels and spring dampers for easy adaptation to ropes of different diameters. Wheeled climbing robots have high speeds, continuous movement, simple structures, simple controls, and low energy consumption; however, their obstacle negotiation ability is weak.

#### 4.3.3. Tracked Locomotion

Tracked climbing robots are inspired by tanks. They have a large contact area, fast speed, continuous movement, and strong obstacle negotiation ability. They are widely used in scenarios where speed, continuous movement, and obstacle negotiation ability are required simultaneously. Cho et al. [40] designed a two-module tracked-type cable-climbing robot called MRC^2^IN-II for the inspection of suspension bridges. The robot consists of two tracks, two safety landing devices, and an attachment device enclosed by bolts. Nguyen et al. [59] developed a roller-chain-like steel-bridge-climbing robot with a tank-like shape for the inspection of municipal steel bridges. The robot consists of two rows of roller chains and a support frame. The robot controls the contact angle between the roller chain and steel bridge via a linear reciprocating drive device that can adapt to bridge surfaces with various shapes. Sun et al. [41] adopted a tracked pole-climbing robot. The drive device is composed of two sets of chain drive mechanisms arranged opposite to each other and a DC deceleration motor. The motor drives the chains on both sides to rotate through the gear transmission, and the robot’s climbing action is realized by the friction between the rubber teeth on the chain and the wire rope. Unver et al. [143] developed a tank-like climbing robot called Tankbot. It weighs only 115 g, can carry 300 g on ordinary painted walls, can cross obstacles up to 16 mm in diameter, and can perform vertical wall-to-ceiling conversions. Liu et al. [144] designed a tracked-type wall-climbing robot named SpinyCrawler. The robot is driven by a roller chain driven by a motor. It can climb on rough walls, such as vertical concrete walls, gravel walls, sandpaper walls, and brick walls, and can also traverse brick ceilings. The disadvantage of tracked climbing robots is that turning is not easy to control.

#### 4.3.4. Legged Locomotion

Legged climbing robots are inspired by the limbs of humans or animals. Legged robots can be divided into three types, series, parallel, and series-parallel hybrid, and may be two-, four-, six-, or multi-legged. For example, the InchwormClimber robot [145] adopts a two-legged climbing structure. The robot consists of two links and three revolute joints. The robot relies on a magnetic wheel to adsorb on a surface, relies on a motor to realize two-legged movement through the belt drive, and completes up-and-down climbing motion according to a certain gait sequence. Parness et al. [146] developed a four-legged climbing robot called LEMUR 3 consisting of a torso, four legs, and four grippers. Each leg has seven degrees of freedom and can freely climb lava caves and solar glass panels in outer space. Bandyopadhyay et al. [147] designed a quadruped climbing robot called Magneto, which consists of 3-DOF actuated limbs and a 3-DOF compliant magnetic foot. It can change its structure and navigate on any slope, as well as through thin beams with different spacings. To increase stability and payload, some climbing robots use six-legged structures, e.g., DIGbot [22] and Abigaille-III [120]. The advantages of legged climbing robots are that they can use a variety of climbing gaits, have strong environmental adaptability, and have a strong ability to overcome obstacles. However, they require complex control systems.

#### 4.3.5. Inchworm Locomotion

As their name suggests, inchworm-style climbing robots are inspired by inchworms. These robots usually consist of two separable parts: one fixed, and one that slides or rotates. They can achieve long-distance climbing tasks. Zheng et al. [26] developed an inchworm-style cable-climbing robot dubbed CCRobot. The robot consists of a clamping module and a parallel operating arm. The clamping module is divided into upper and lower parts. The parallel operation arm consists of upper and lower platforms and three sets of 3-RPS (Revolute–Prismatic–Spherical) articulated arms. The upper platform and upper arm are connected by a ball joint, the lower platform and lower arm are connected by a rotating joint, and the upper and lower arms are connected by a moving joint, and are all driven by a DC motor. Sun et al. [148] designed an inchworm-style climbing robot for cleaning the glass on high-rise buildings. The robot consists of two mutually perpendicular rodless cylinders, a rotary cylinder, four *Z*-axis lift cylinders, and sixteen suction cups. The suction cups stick to the glass and autonomous climbing is achieved through the alternating rotation of two rodless cylinders. The advantages of inchworm-type climbing robots are that their structures and controls are relatively simple. Their disadvantages are discontinuous movement and slow speed.

#### 4.3.6. Hybrid Locomotion

Hybrid climbing robots combine the advantages of two or more forms of climbing structures and can adapt to more complex climbing environments. Mguyen et al. [58] designed a wheel-leg hybrid steel-bridge-climbing robot consisting of a torso and two legs. When moving on a flat surface, the two legs are fixed in position and mainly move in a wheeled manner. When crossing obstacles, one of the legs is fixed and the other leg can be extended to move in a walking manner. The pipe-climbing robot designed by Han et al. [13] adopts a 4-DOF wheel-leg climbing structure. The robot consists of two drive modules and a connecting arm. On smooth pipes, the robot uses a wheeled climbing mode to move quickly. When it needs to overcome obstacles such as elbows or T-joints, it switches to a legged climbing mode. Moon et al. [149] used a combination of the rope-driven mode and guide-rail mode in a maintenance robot system to allow it to climb the facades of high-rise buildings.

Hybrid climbing robots have strong environmental adaptability and good comprehensive performance; however, they require a relatively complex structure. The advantages, disadvantages, and performance of the various locomotion modes are compared in Table 2.

### 4.4. Security Mechanisms

In an emergency such as a sudden power failure, it is critical that a climbing robot not fall from the climbed object. So, devices for safe landing and recovery are required. Most climbing robots are driven by DC gear motors or servo motors. Such motors often have a self-locking mechanism such as a worm gear, which acts as a safety feature in the event of a power failure. In addition, some robots use special safe-landing devices. WRC^2^IN-II [40] adopts a safe-landing device composed of a timing belt, a pulley, spur gears, a disc damper, a reverse braking device, and a support shaft. When the robot descends due to a loss of power, the synchronous belt drives the pulley, spur gear, and internal device of the disc damper to rotate. The reverse braking device is fixed when there is a loss of power, so that the external device of the disc damper is fixed. The disc damper contains viscous silicone oil that damps energy during the robot’s descent, so that it can land safely. If the robot gets stuck on the cable, the robot can use a clutch mechanism to allow it to return safely to the ground. Xu et al. [33] used a gas-damper safe-landing device with a sliding rod mechanism in a cable-climbing robot. The safe-landing mechanism consists of a cylinder and a slider mechanism. A crank is fixed to the driveshaft by a one-way clutch. When the robot climbs, the one-way clutch is released. As the robot slides down, a drive wheel drives the slider–crank mechanism via the clutch. The rotational motion of the drive wheel is converted into reciprocating motion of the piston in the cylinder. The gas in the cylinder is alternately inhaled or discharged through nozzles arranged on the bottom wall of the cylinder, forming a gas damper that consumes the kinetic energy of the robot. The size of the nozzle can be adjusted to obtain different damping rates to control the landing speed of the robot. Gui et al. [150] used active and passive anti-fall devices in a tree-pruning robot to prevent it from falling to the ground. The passive anti-falling mechanism uses only friction forces and robot gravity force to maintain a hold on the tree trunk. The active anti-fall mechanism adjusts the distance between the wheel and trunk using a stepper motor and a lead screw nut unit. While safe-landing and recovery devices can improve the safety factors of robots by ensuring that they can be recovered after a power failure, they also increase their weight.

### 4.5. Control Methods

The control system is the core of a climbing robot. Its main task is to control the robot’s actuator to complete specified movements and functions according to the user’s work instructions, the robot’s control programs, or feedback from sensors. The control system mostly adopts a master–slave system composed of two parts: a ground monitoring station and a robot controller. The ground station is usually composed of a portable PC, smartphone, remote control, and game handle. The robot controller mostly comprises microprocessors or single-board computers such as an Arduino, Raspberry Pi, MCS51, PIC, STM32, or PLC. Because climbing robots often need to travel long distances, the ground station and robot controller often use wireless communication methods such as Bluetooth and WiFi. A few robots with short travel distances directly use RS232 or USB for communication. Tavakoli et al. [103] used a wired control system in the 3DClimber robot. The control system consists of a host computer and a controller, which are connected through USB and can send commands and receive sensor information. The controller adopts the CANopen protocol for communication, and controls the position, speed, and torque of each AC or DC motor. Sun et al. [41] adopted a two-layer wireless control system in a pole-climbing robot. The control system consists of an STM32 microprocessor, a wireless signal transmitter, and a wireless graphic transmitter. The WRC^2^IN cable-detection robots adopt a three-layer wireless control system that consists of a remote portable visual monitoring platform, a master controller, and a slave controller [37]. The monitoring platform is used to issue commands and receive display information. The master controller is used to store the robot’s pose state and sensor information and communicate with the monitoring platform. The slave controller is composed of a single-board computer (SBC), which is used to control the motors. The monitoring platform and master controller use the Xbee mode to communicate wirelessly, and the master and slave controllers use the CAN bus mode to communicate. The Rise-Rover climbing robot adopts a three-layer wireless control strategy which consists of a user layer, a middle layer, and a bottom layer [45]. The user layer is an Android mobile phone platform, which is mainly used as a user interface for remote control and video monitoring. The middle layer is an embedded Linux platform, which mainly handles peripheral devices, such as cameras and NDT devices. The bottom layer is controlled by an Arduino controller, which mainly deals with real-time control of the motor and PID control of air pressure.

Apart from the hardware components, some robots also use software to realize a human–machine interface and improve the robot’s autonomy. In the Waalbot II robot [117], a two-level motion planner is implemented, so that transitions between locally flat regions are identified using the upper planner and the specific robot trajectory is planned using an A* search algorithm. To implement autonomous climbing in the Climbot robot [10], a truss modeling and recognition system has been proposed. The system adopts a Truss Segmentation Pouring Algorithm and a Truss Parametric Expression Algorithm to recognize truss-style structures. Li et al. [151] developed a robotic system for the automatic inspection of weld defects in spherical tanks. The robot adopts a weld-line tracking method based on deep learning, as well as an optimal path-planning method for traversing all the weld lines of a spherical tank.

### 4.6. Operating Tools

Climbing robots are mainly used to carry tools to conduct various tasks, such as inspection, cleaning, spraying, welding, maintenance, and pruning. These tools may include cameras, manipulators, nondestructive testing equipment, laser-cleaning equipment, and spraying equipment. Some work tools are off-the-shelf, while others require customization. The designers must consider how these tools are carried by the robot and their impact on climbing performance. Xu, et al. [36] installed two cameras and a nondestructive testing device on a cable-climbing robot to carry out cable inspection. Cho et al. [40] installed four cameras and a nondestructive testing device in the MRC^2^IN suspension bridge cable inspection robot. In the glass curtain wall inspection robot developed by Liang et al. [152], an operating arm is used to detect the firmness of the glass installation. Huang et al. [153] designed a multifunctional pruning and crushing end effector for automatic pruning of fruit trees. The Model-IV cable maintenance robot [36] uses four working modules for grinding, cleaning, spraying, and winding. Lee et al. [53] used a window-cleaning device in a wall-climbing cleaning robot. Tools will increase the weight of a robot and change its center of gravity, which will affect its climbing performance.

## 5. Typical Climbing Robots

In the past decade, a large number of climbing robots have been developed. Table 3 presents a list of some typical climbing robots according to the above-mentioned classes. Some typical robotic prototypes without specific names are not listed.

## 6. Challenges and Future Research Directions in Climbing Robots

### 6.1. Challenges Faced

After decades of development, climbing robots have made great progress in terms of adhesion, locomotion, and control methods. However, there are very few climbing robots that have been widely used in the market. The main reason for this is that there are still many unresolved problems and challenges in the development of this technology, as described below:

(1) Multi-environmental adaptation problems. Due to the variety of climbed objects, no climbing robot can achieve stable climbing and complete tasks in various complex unstructured environments.

(2) Application problems. Although hundreds of prototypes of climbing robots have been developed around the world, most of them are still in the laboratory research stage and cannot be adapted to complex industrial and agricultural field environments. Most climbing robots are only equipped with cameras and a few sensors and lack other working tools, which limits their application scope.

(3) Energy supply problems. As a cable-powered robot gains height, the length of the cable increases and, hence, so does its overall weight. Battery-powered robots have a limited life, so continuous research is required to improve battery life.

(4) The issue of autonomy. Most current climbing robots can only work under manual or semi-automatic conditions. It is difficult to achieve autonomous operation due to the complexity of the environment.

### 6.2. Main Future Research Directions

With the development of new materials and technologies, future research on climbing robots will focus on improving the reliability of adhesion mechanisms, the operability and autonomy of movement, and the development of related operating tools. The main aspects are as follows:

(1) Bionic climbing robots. Bionic design is widely used in product design, architectural design, and other fields. Many animals have strong adhesion and climbing abilities, providing a good reference for research on climbing robots. Researchers study the shape, structure, and function of animals and apply this knowledge to robot design via mathematical modeling, mechanical analysis, digital simulation, virtual simulation, and other means. Bionic design for climbing robots focuses on new bionic materials, bionic mechanisms, attachment methods, and the imitation of gaits.

(2) Modular climbing robots. The modular method is a basic way to solve complex problems. It combines simple modules to form a complex system that is universal, reconfigurable, extensible, and self-healing. As well as their low cost, they are widely used in the development of complex electromechanical systems, such as automated assembly lines and robots. Through modular design, a climbing robot system can be constructed with many of the same or different adhesion modules, motion modules, and control modules. These modules are independent and complete units that can be easily connected or disconnected from each other; thereby, robotic systems with many different purposes and functions can be built.

(3) Intelligent climbing robots. Intelligence means giving robots certain human behaviors and cognitive and decision-making functions so that they can respond autonomously to changes in the surrounding environment. The intelligent design of climbing robots mainly focuses on intelligent control. With the help of various sensors, as well as machine vision, deep learning, and other technologies, a robot can autonomously identify the surrounding environment, automatically plan a movement path, and autonomously cross obstacles.

(4) Lightweight designs. The weight of the robot directly affects its climbing and loading performance, so it should be minimized. The main idea of lightweight design is to use lightweight materials such as high-strength steel, aluminum alloy, carbon fiber, and engineering plastics. Another method is to optimize the structure of the robot through finite element analysis.

(5) Flexible and soft climbing robots. Compared with rigid climbing robots, flexible and soft climbing robots have better environmental adaptability, safety, and human–computer interaction capabilities. Future research on flexible climbing robots will mainly focus on the utilization of flexible materials such as liquid silicone rubber, hydrogels, electroactive polymers, shape memory alloys, shape memory polymers, and liquid metals, as well as liquid actuators.

(6) Hybrid designs. At present, a variety of adhesion and locomotion methods have been developed for climbing robots, each with its own advantages, disadvantages, and adaptability. One main research direction is the hybrid design of multiple adhesion and locomotion methods, so that climbing robots can adapt to complex environments.

(7) Integrated design. Integrated design is a common design method used to improve productivity. The integrated design of climbing robots integrates adhesion devices, mobile devices, control platforms, and operating tools to allow them to complete certain climbing tasks.

(8) Multi-machine collaboration. In large-scale operating environments, relying on single robots is no longer possible. The use of multiple robots can also enhance flexibility, especially in the optimization of resource allocation and scheduling. Research on multi-robot synergy focuses on collaborative perception, collaborative planning, and collaborative control.

## 7. Conclusions

Climbing robots have good application potential in scenarios that are difficult or dangerous for humans to work in. This paper reviewed the past decade’s research on bionic climbing robots designed for climbing vertical structures such as poles, cables, walls, and trees, and discussed some of their applications. Some key aspects, such as conceptual design, adhesion mechanisms, locomotion modes, safety mechanisms, control methods, and operating tools, were explained using examples. The advantages, disadvantages, and applications of each method were compared and analyzed. Finally, the challenges faced by climbing robots and the main future research directions were discussed.

## Figures and Tables

**Figure 1 biomimetics-08-00047-f001:**
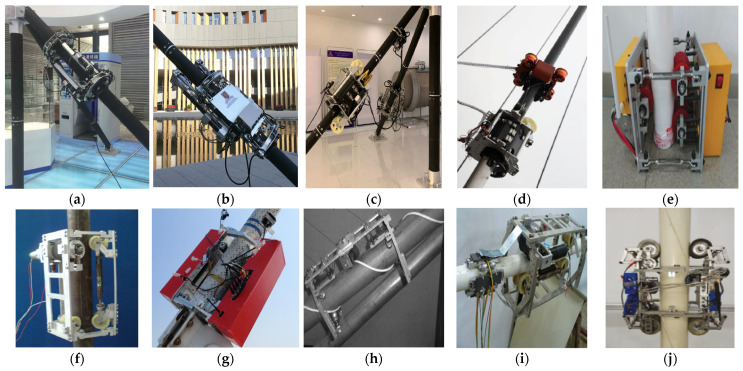
Cable-climbing robots. (**a**) CCRobot-I [Reproduced from [26] with permission from Ning Ding]; (**b**) CCRobot-II [Reproduced from [27] with permission from Ning Ding]; (**c**) CCRobot-III [Reproduced from [28] with permission from Ning Ding]; (**d**) CCRobot-IV [Reproduced from [29] with permission from Ning Ding]; (**e**) Robot [Reproduced from [31] with permission from Ning Ding Bin He]; (**f**) Model-I [Reproduced from [32] with permission from Fengyu Xu]; (**g**) Model-II [Reproduced from [33] with permission from Fengyu Xu]; (**h**) Model-III [Reproduced from [34] with permission from Fengyu Xu]; (**i**) Robot [Reproduced from [35] with permission from Fengyu Xu]; (**j**) Model-IV [Reproduced from [36] with permission from Fengyu Xu].

**Figure 2 biomimetics-08-00047-f002:**
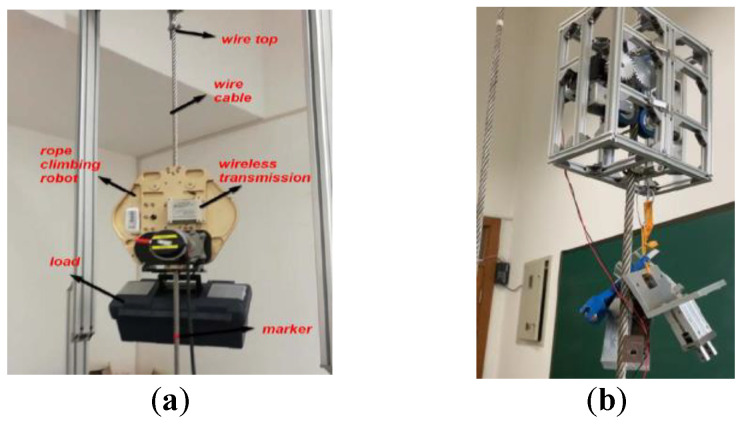
Rope-climbing robots. (**a**) Robot [Reproduced from [41] with permission from Guanglin Sun]; (**b**) WRR-II [Reproduced from [43] with permission from Guisheng Fang].

**Figure 3 biomimetics-08-00047-f003:**
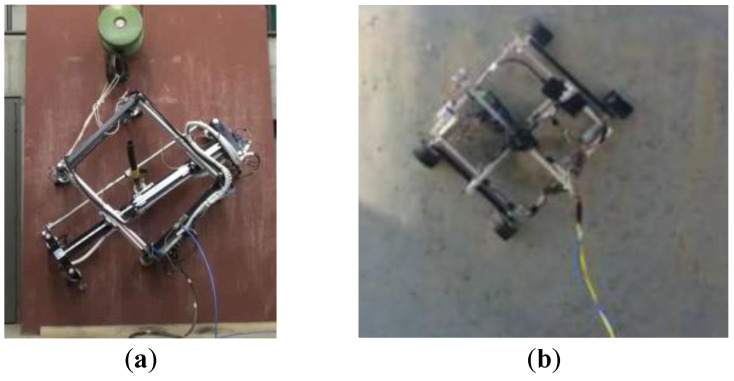
Wall-climbing robots.; (**a**) sandblasting robot [Reproduced from [51] with permission from Richard J. Duro]; (**b**) EJBot [Reproduced from [52] with permission from Mohamed Gouda Alkalla].

**Figure 4 biomimetics-08-00047-f004:**
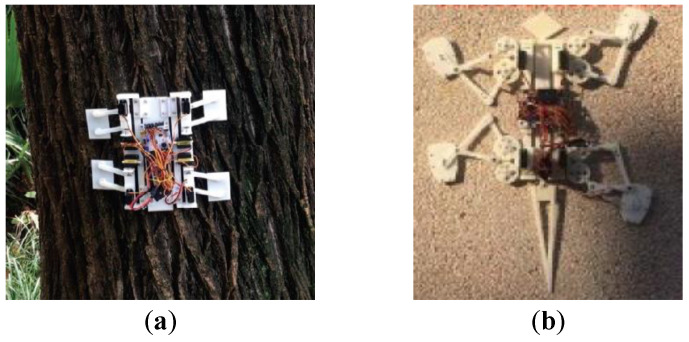
Bionic climbing robots. (**a**) longicorn-like robot [Reproduced from [73] with permission from Deyi Kong]; (**b**) cicada- and gecko-like robot [Reproduced from [74] with permission from Deyi Kong].

**Table 1 biomimetics-08-00047-t001:** Comparison of various robot adhesion methods.

Method	Advantages	Disadvantages	Applicable Scope	Representative Products
Magnetic adsorption	Large adsorption force, permanent magnet does not need electricity	Magnets are generally heavy, which increases weight and reduces the load capacity	Ferromagnetic materials	Omniclimbers, Minibobot-W
Air pressure adsorption	Large adsorption force, easy to control, regardless of materials	High energy consumption, noise, large size, movement delay, poor safety	Flat, smooth non-porous and non-cracked surfaces	W-Climbot, Vortexbot, EJBot, UOTWCR, Rise-Rover
Clamping adhesion	Low energy consumption, no noise, strong load capacity	Clamping directivity	Slender objects such as rods or tubes	3DClimber, WRC^2^IN
Claw grasping	No energy consumption, no noise, strong load capacity	Damages soft objects	Rough objects with bulges or pits	Spinybot, DynoClimber, Tbot, Treebot LEMUR
Adhesive adsorption	No energy consumption, no noise	Weak load capacity and slow movement speed	Smooth objects	Abigaille-III, AnyClimb, Waalbot ThermsBond
Electrostatic adsorption	Low weight, small dimensions, low energy consumption, and no noise	Low load capacity, slow speed, sensitivity to surface conditions involving dust	Uncontaminated and uncharged objects	[128,129]
Hybrid adhesion	Good comprehensive performance	Complex structure	Adapts to a variety of environments	[132,133,134]

**Table 2 biomimetics-08-00047-t002:** Comparison of locomotion performance.

Locomotion Mode	Advantages	Disadvantages	Applicable Scope	Representative Robots
Rope-driven	Strong load-carrying capacity, fast, high degrees of stability and safety	Requires a winch, limited movement, high manufacturing and installation costs	Scenarios requiring heavy loads	PiSaRo2
Wheeled	Fast, continuous movement, simple structure, simple control, low energy consumption	Weak obstacle- negotiation ability	Flat objects	WRC^2^IN-I UT-PCR WCR-Eto
Tracked	Large contact area, fast, continuous movement, strong obstacle-climbing ability	Complex structures, difficulty turning	Scenarios with obstacles	Tankbot, SpinyCrawler, MultiTank, Rise-Rover
Legged	Environmental adaptability, ability to overcome obstacles	Complex structure, complex control, discontinuous movement, slow	Scenarios with substantial obstacles	InchwormClimber, DIGbot Climbot
Inchworm	Simple structure, simple control, high safety factor	Discontinuous movement, slow	Scenarios with small obstacles	CCRobot, CROC, Treebot, Pylon-Climber, EJBot
Hybrid	Environmental adaptability, good comprehensive performance	Complex structure	Complex climbing environments	OmniClimber

**Table 3 biomimetics-08-00047-t003:** List of climbing and operating robots.

Robot Name	Category	Adhesion	Locomotion	Controller	Tools	Country	Year
UT-PCR	Pole-climbing	Clamping	Wheeled	Unknown	Camera, washing devices	IR	2011
Climbot	Pole-climbing	Clamping	Legged	Accelnet	Grippers, Camera	CHN	2011
EVOC-1	Pole-climbing	Clamping	Inchworm	Unknown	Unknown	CHN	2019
Snake-like robot	Pole-climbing	Clamping	Inchworm	Arduino	Unknown	CHN	2020
DIGbot	Tree-climbing	Claw	Legged	SBC	Unknown	US	2010
Treebot	Tree-climbing	Claw	Inchworm	Unknown	Unknown	CHN	2011
PylonClimber-I	Pylon-climbing	Clamping	Inchworm	C8051	Unknown	CHN	2017
PylonClimber-II	Pylon-climbing	Clamping	Inchworm	C8051	Unknown	CHN	2018
CROC	Bridge-climbing	Magnetic	Inchworm	Unknown	Unknown	AUS	2014
ARA-I robot	Bridge-climbing	Magnetic	Tracked	Unknown	Camera	US	2019
ARA-II robot	Bridge-climbing	Magnetic	Hybrid	Arduino	Unknown	US	2020
WCR^2^IN-I	Cable-climbing	Clamping	Wheeled	SBC	Camera, NDT	KR	2012
WCR^2^IN-II	Cable-climbing	Clamping	Tracked	SBC	Camera, NDT	KR	2014
EJBot	Cable-climbing	Pressure	Wheeled	Arduino	Camera	EGY	2017
CCRobot-I	Cable-climbing	Clamping	Inchworm	STM32	Camera	CHN	2018
CCRobot-II	Cable-climbing	Clamping	Inchworm	STM32	Camera	CHN	2019
CCRobot-III	Cable-climbing	Clamping	Hybrid	SoC	Camera	CHN	2020
CCRobot-IV	Cable-climbing	Clamping	Hybrid	PX4	Camera	CHN	2021
Model-1	Cable-climbing	Clamping	Wheeled	STM32	Camera, NDT	CHN	2012
Model-2	Cable-climbing	Clamping	Wheeled	STM32	Camera, NDT	CHN	2014
Model-3	Cable-climbing	Clamping	Wheeled	STM32	Camera, NDT	CHN	2015
Model-4	Cable-climbing	Clamping	Hybrid	STM32	Grinding devices	CHN	2021
Waalbot II	Wall-climbing	Adhesive	Hybrid	VICON	Camera	US	2011
Minibobot-W	Wall-climbing	Magnetic	Inchworm	C8051	Probe	CHN	2012
W-Climbot	Wall-climbing	Pressure	Legged	Accelnet	Camera	CHN	2012
MultiTank	Wall-climbing	Pressure	Tracked	PIC	Unknown	KR	2013
LARVA-II	Wall-climbing	Pressure	Wheeled	Unknown	Camera	KR	2013
Abigaille-II	Wall-climbing	Adhesive	Legged	FPGA	Unknown	CAN	2012
Abigaille-III	Wall-climbing	Adhesive	Legged	FPGA	Unknown	CAN	2014
ACROBOT	Wall-climbing	Adhesive	Inchworm	Baby orangutan	Unknown	US	2014
Rise-Rover	Wall-climbing	Pressure	Tracked	PIC	NDT	USA	2015
Tbot	Wall-climbing	Claw	Wheeled	Unknown	Unknown	CHN	2015
OmniClimber-I	Wall-climbing	Magnetic	Hybrid	STM32	Unknown	PT	2014
OmniClimber-II	Wall-climbing	Magnetic	Hybrid	STM32	Unknown	PT	2016
MARC	Wall-climbing	Magnetic	Tracked	Unknown	Camera	ITA	2017
Vortexbot	Wall-climbing	Pressure	Wheeled	Arduino	Unknown	CHN	2017
LEMUR 3	Wall-climbing	Claw/Adhesive	Legged	VDX-6354	Unknown	US	2017
PiSaRo2	Wall-climbing	No	Wire-driven	RPi	Unknown	FR	2018
AnyClimb-I	Wall-climbing	Adhesive	Inchworm	Unknown	Unknown	KR	2016
AnyClimb-II	Wall-climbing	Adhesive	Inchworm	Unknown	Unknown	KR	2018
Mantis	Wall-climbing	Pressure	Tracked	Arduino	Unknown	SG	2019
UOTWCR-II	Wall-climbing	Pressure	Wheeled	Unknown	Unknown	IRQ	2020
SpinyCrawler	Wall-climbing	Claw	Tracked	Unknown	Unknown	CHN	2020
Ibex	Wall-climbing	Pressure	Wheeled	Arduino	Unknown	SG	2020
GFCR	Wall-climbing	Pressure	Hybrid	Arduino	Roller brush	IN	2022
Clothbot	Cloth-climbing	Claw	Wheeled	Unknown	Unknown	CHN	2012
LEeCH	Various applications	Pressure	Inchworm	Arduino	Unknown	JPN	2019

## Data Availability

Not applicable.

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
