# Peer review of "Advances in Climbing Robots for Vertical Structures in the Past Decade: A Review"

_biomimetics, 2023, doi:10.3390/biomimetics8010047_

Round 1

Reviewer 1 Report

This manuscript deals with a lot of climbing robots as the review paper. It is required to revise the manuscript according to comments:

l  It seems that there is a considerable difference between the tile and the content of the manuscript in two points. Even though the title includes “recent advances”, it does not mention recent publications. The manuscript with “bionic climbing robots” in the title include many non-bionic or non-bioinspired robots.

l  Several review papers on climbing robots have been published, for example, recent paper by Fang et al. [1]. Therefore, it is suggested the authors focus on bionic or bioinspired robots to fit the main topic. The biomimetic research in section 4.1 should be expanded by adding recent research. For example, additional animal-inspired climbing robots (such as [2]), plant-inspired robots (publications mentioned in section 5 of [3] and [4]) and bio-inspired adhesive applications [5].

[1]   Fang et al. Design and technical development of wall-climbing robots: A review. Journal of Bionic Engineering. 2022, 19, 877-901

[2]   Han et al., A miniaturized wall-climbing segment robot inspired by caterpillar locomotion, 2017, Bioinspir. Biomim. 12, 046003

[3]   Mazzolai, et al. The bio-engineering approach for plant investigations and growing robots. A mini-review, Front. Robot. AI, 2020, 7, 573014

[4]   Fiorello, et al. Taking inspiration from climbing plants: methodologies and benchmarks—a review. 2020, Bioinspir. Biomimet. 15:031001

[5]   Borijindakul et al. Mini Review: Comparison of Bio-Inspired Adhesive Feet of Climbing Robots on Smooth Vertical Surfaces, 2021, Frontiers in Bioengineering and Biotechnology, 9, 765718

Author Response

Dear reviewer:

   Thank you very much for your suggestions. I deeply appreciate the time and effort you’ve spent in reviewing our manuscript. Your comments are really thoughtful and helpful. We revised the manuscript, following your comments exactly. 

Point 1:  It seems that there is a considerable difference between the tile and the content of the manuscript in two points. Even though the title includes “recent advances”, it does not mention recent publications. The manuscript with “bionic climbing robots” in the title include many non-bionic or non-bioinspired robots.

Response 1: The content of the article included most bionic climbing robots and some typical non-bioinspired climbing robots developed in the past decade, so the title of the article has been revised to “Advances in Climbing Robots for Vertical Structures in the Past Decade: A Review”.

Point 2: Several review papers on climbing robots have been published, for example, recent paper by Fang et al. [1]. Therefore, it is suggested the authors focus on bionic or bioinspired robots to fit the main topic. The biomimetic research in section 4.1 should be expanded by adding recent research. For example, additional animal-inspired climbing robots (such as [2]), plant-inspired robots (publications mentioned in section 5 of [3] and [4]) and bio-inspired adhesive applications [5].

Response 2: All of the references that you mentioned above have been added in the revised article.

Point 3: Moderate English changes required.

Response 3: The article has been checked and corrected for proper English language, grammar, punctuation, spelling, and overall style by one or more of the highly-qualified, native English speaking editors at Native English Editing.

Thanks and Best regards!

Yours sincerely

Reviewer 2 Report

This paper analyzes and compares the past decade’s design and development of bionic climbing robots that can ascend vertical structures such as rods, cables, walls, and trees.

Firstly, the main research fields and basic design requirements of climbing robots are introduced, then the advantages and disadvantages of six key technologies are summarized, namely, conceptual design, adhesion methods, locomotion modes, safety mechanisms, control methods, and operational tools.

This paper provides a scientific reference for researchers engaged in the study of climbing robots.

This paper reviewed the past decade’s research into bionic climbing robots designed for climbing vertical structures such as poles, cables, walls, and trees, and discussed some of their applications. Some key aspects, such as conceptual design, adhesion mechanisms, locomotion modes, safety mechanisms, control methods, and operating tools, were explained using examples. The advantages, disadvantages, and applications of each method were compared and analyzed.

Finally, the challenges faced by climbing robots and the main future research directions were discussed.

It is clear from the presented contribution that it will be necessary to continue the research.

I consider the presented article to be original and it contains a considerable amount of used literature. The current scientific literature is used in the article.

In the submitted article, the mathematical apparatus could have been applied to a greater extent.

The methodology of the article is appropriately applied in the submitted contribution.

The paper is well structured. Conclusions are clear, in line with the main text. The manuscript is clear written and could be interesting for the researchers from this field.

All the best!   

Author Response

Dear reviewer:

     Thank you very much for your support!

Yours sincerely.

Round 2

Reviewer 1 Report

The manuscript has been revised based on the reviewer’s comments. I would recommend the manuscript for publication after the minor revision according to the following comment.

[1]   The authors include only the recent papers on climbing robots that the reviewer mentioned in the reviewer report. It is suggested that an important paper published recently as a cover study in Science Robotics:

Agile and versatile climbing on ferromagnetic surfaces with a quadrupedal robot, December 14 2022, Science Robotics, Vol 7.

https://www.science.org/doi/10.1126/scirobotics.add1017